# Strengthening provider accountability: A scoping review of accountability/monitoring frameworks for quality of RMNCH care

Eva Jarawan[1], Mara Boiangiu[2], Wu Zeng[1]*

**1** Department of Global Health, School of Health, Georgetown University, Washington, DC, United States of America, **2** Georgetown University, Washington, DC, United States of America

* wz192@georgetown.edu

**Data Availability Statement:** All indicators under review are publicly available. Data on indicators used in the QED framework is obtained available from https://cdn.who.int/media/docs/default-source/mca-documents/qoc/qed-quality-of-care-

## Abstract

Increasing health providers' accountability is an important element in improving quality of care (QoC) for reproductive, maternal, neonatal, and child health (RMNCH), so as to improve health outcomes of the population in many low- and middle-income countries (LMICs). Implemented RMNCH monitoring initiatives vary in their settings, methods of data collection, and indicators selected for monitoring. The purpose of this study is to evaluate the monitoring/accountability frameworks used by key global monitoring initiatives and provide insights for countries to develop context-customized indicators for RMNCH monitoring and accountability in middle-income countries. The authors conducted a scoping review of key global monitoring initiatives on their monitoring/accountability framework and associated indicators. Data was extracted into a spreadsheet template for analysis. Monitoring/accountability frameworks corresponding to the selected global RMNCH initiatives were described, analyzed, and then categorized the monitoring indicators used by the initiatives according to the type of indicators, quality domains, monitoring levels, and type of services. The results showed that all frameworks regarded developing quality indicators and their monitoring as important elements of accountability and emphasized the role of health systems blocks as inputs for QoC. The researchers demonstrated the importance of measuring quality through both condition-specific and general health system indicators. However, given the different purposes of global monitoring initiatives, the indicators they used varied. We found a lack of indicators measuring QoC of reproductive health. In terms of quality domains, the timeliness and efficiency of RMNCH services were neglected, as few of these indicators were selected for monitoring. Global monitoring initiatives provide valuable frameworks for countries to understand which key indicators need to be tracked to achieve global objectives and develop the foundation for their own accountability/monitoring systems. Gaps in quality indicator design and use emphasize countries need to build on what the global initiatives have achieved to systematically examine quality concerns, develop a tailored and effective accountability/monitoring framework, and improve population health.

for-maternal-and-newborn-health-a-monitoring-framework-for-network-countries.pdf?sfvrsn=19a9f7d0_1. Data on indicators used in Countdown to 2030 is obtained from https://www.countdown2030.org/wp-content/uploads/2021/12/Countdown-to-2030-2021-Annex_Nov_23_2021.pdf. Data on indicators used in Global Strategy is obtained from https://data.unicef.org/resources/indicator-monitoring-framework-global-strategy-womens-childrens-adolescents-health/. Data on Core 100 Health Indicators is obtained from https://score.tools.who.int/fileadmin/uploads/score/Documents/Enable_data_use_for_policy_and_action/100_Core_Health_Indicators_2018.pdf. The authors did not have any special access privileges that other would not have.

**Funding:** Funding support is from Georgetown University Medical Center for publication fees. The authors received no specific funding for the study itself.

**Competing interests:** The authors have declared that no competing interests exist.

## Introduction

Improving quality of care has become increasingly important as part of low- and middle-income countries' (LMICs) efforts to achieve the Sustainable Development Goals (SDGs). In recent decades, we have observed significant improvement in the coverage of reproductive, maternal, newborn, and child health (RMNCH) services [1]. Ensuring and improving quality of care (QoC) becomes more and more critical to saving lives [2].

Many strategies have been developed and implemented to improve QoC. The strategies include–but are not limited to–training and coaching, accreditation, public reporting, performance-based payment, and continuous quality improvement (QI) initiatives [3, 4]. Although the effectiveness of these approaches on QoC varies because they do not work universally across all settings, there is a consensus that accountability is the core element embedded in all approaches to improving QoC.

Accountability is not a new concept in the health setting. It is commonly referred to as the mechanism under which a governing body is able to mandate providers or organizations to meet certain goals or objectives [5], and emphasizes the indicators that are set for providers or organizations to achieve. The indicators can be used to hold health providers accountable for what they produce by comparing their performances and monitoring progress towards certain goals. Accountability in healthcare is often concerned with setting up regulations to stipulate: (a) to whom health workers are held accountable; (b) which activities they are accountable for; and (c) the procedures by which their accountability is evaluated [6]. The concept of accountability has been widely used for quality of care and patient safety [4]. In fact, accountability is regarded as an important and integrated element in supporting continuous quality improvement (CQI) in healthcare [7, 8], as one of the steps in CQI is to assess where measured outcomes meet the expectation (check) in CQI's Plan-Do-Check-Act (PDCA) cycle. Thus, it requires the development of valid accountability measures to guide quality improvement [9]. Given that accountability for QoC substantially overlaps with the effort of monitoring quality indicators, we use the terms "accountability framework" and "monitoring framework" interchangeably in this paper.

Understanding existing common frameworks helps countries to develop their own accountability frameworks to improve or ensure a certain level of QoC for RMNCH services. This study aims to review and describe a few existing monitoring frameworks for RMNCH services, analyze their strengths and limitations, and identify gaps for further improvement. This information could guide countries to develop their own accountability/monitoring frameworks for improving QoC.

## Methods

We used a scoping review approach to review existing global monitoring frameworks on RMNCH indicators. Because the main purpose of this study is to gauge the adequacy and complexity of monitoring indicators measuring the quality of care for maternal and child services at the global level and to identify knowledge gaps, scoping reviews were recommended and used in this study [10]. There were several accountability/monitoring frameworks that are widely used in the global setting. We included them into the review, incorporating theoretical frameworks and associated indicators.

### Search approach

We used a purposive approach to identify potential frameworks proposed by global monitoring initiatives for RMNCH services for the review. Primarily informed by prior studies [11, 12], we selected the following global monitoring initiatives that focused on monitoring

RMNCH services for the review: (a) WHO's quality, equity, and dignity (QED) network [13]; (b) Countdown to 2030 [14]; (c) The Global Strategy for Women, Children's, and Adolescents' Health (Every Woman Every Child [EWEC] Network) [15]; and (d) Global Reference List of 100 core health indicators [16]. For example, nine global monitoring initiatives related to RMNCH services were identified by Moller [12] while Hilber et al. leveraged the Global Strategy for Women, Children's, Adolescents' Health to propose potential refinement of the accountability measurement framework for global health initiatives [11]. The following criteria were applied to select specific global monitoring initiatives to review: (a) the initiative should primarily concern maternal and/or child health; (b) the initiative concerns developing or using monitoring indicators for maternal and/or child health; (c) the monitoring indicators should contain indicators measuring the quality of care for maternal and/or child health; and (d) the initiative, along with its monitoring framework, should be applied to multiple countries and/or to the global level. The search was initially conducted on July 13, 2022 and updated on November 20, 2022. A more detailed description of each initiative can be found in S1 Appendix.

For initiatives (b)-(d) mentioned above, not all the indicators were related to QoC. To ensure the relevance of the indicators to QoC of RMNCH, we reviewed and extracted RMNCH indicators. If not related to QoC, they were excluded from the analysis. More detailed exclusion criteria are explained in the corresponding section below.

Additionally, we reviewed other global RMNCH initiatives mentioned by Moller and Hilber [11, 12], such as ending preventable maternal mortality (EPMM), and found that EPMM mostly focused on health system improvement with few indicators on QoC of RMNCH services [17]. Thus, it was excluded from the review. We also conducted a search through the Google Scholar and PubMed databases using a combination of the following keywords: "global initiatives", "RMNCH", "accountability" and "monitoring." The search did not yield any additional sources with a focus on QoC for RMNCH services that met the inclusion criteria.

## Review process and inclusion and exclusion criteria

All monitoring indicators from the frameworks were extracted and compiled into an Excel file. The final structure of the database covered key characteristics of the indicators, including indicator names; levels of monitoring (facility level vs. district/regional); types of quality indicators (input, output/process, or outcome); conditions for indicators (reproductive, maternal, newborn, or childcare, or general indicators); possible data sources; and domain of quality of care (safety, effectiveness, efficiency, timeliness, or patient-centered). General indicators were those that did not target a specific population, such as the availability of medication. Regarding the type of quality indicators, we followed Donabedian's classification and categorized them into input, output/process, and outcome indicators [18].

We used the Institute of Medicine's (IOM) STEEEP framework to define the six domains of quality, including safety, effectiveness, timeliness, efficiency, equity, people-centeredness [19]. The definition of each domain is provided in S2 Appendix.

The domain of equity crosses all other domains, and equity indicators are generally derived from other domains' indicators, so this domain was omitted from the analysis. RMNCH indicators not quality-related or indicators not related to RMNCH services were excluded from the analysis. We also excluded indicators that solely measured coverage without the quality element.

## Data analysis

We first described the core elements of the accountability/monitoring framework of each global initiative where the framework was available. All four initiatives, except for the global

reference list of 100 core health indicators, had theoretical frameworks. The descriptions focused on the common themes and the differences between the three frameworks.

For the accountability/monitoring indicators, we developed a classification system to group the indicators. The classification was based on (a) which types of indicators were addressed; (b) domains of quality, according to the IOM's STEEEP framework; (c) services for which the indicators are used; (d) monitoring levels, if available; and (e) potential data sources, if possible. Because the QED framework has the most comprehensive list of indicators, we singled it out when analyzing indicators. We compiled non-duplicated RMNCH indicators from the Countdown to 2030, Global List of 100 core health indicators, and EWEC, then conducted a frequency analysis.

# Results

## Monitoring theoretical framework

Fig 1 shows an adaptation of the accountability/monitoring framework of WHO's QED network [13]. It is one of the most comprehensive frameworks used for monitoring QoC. This

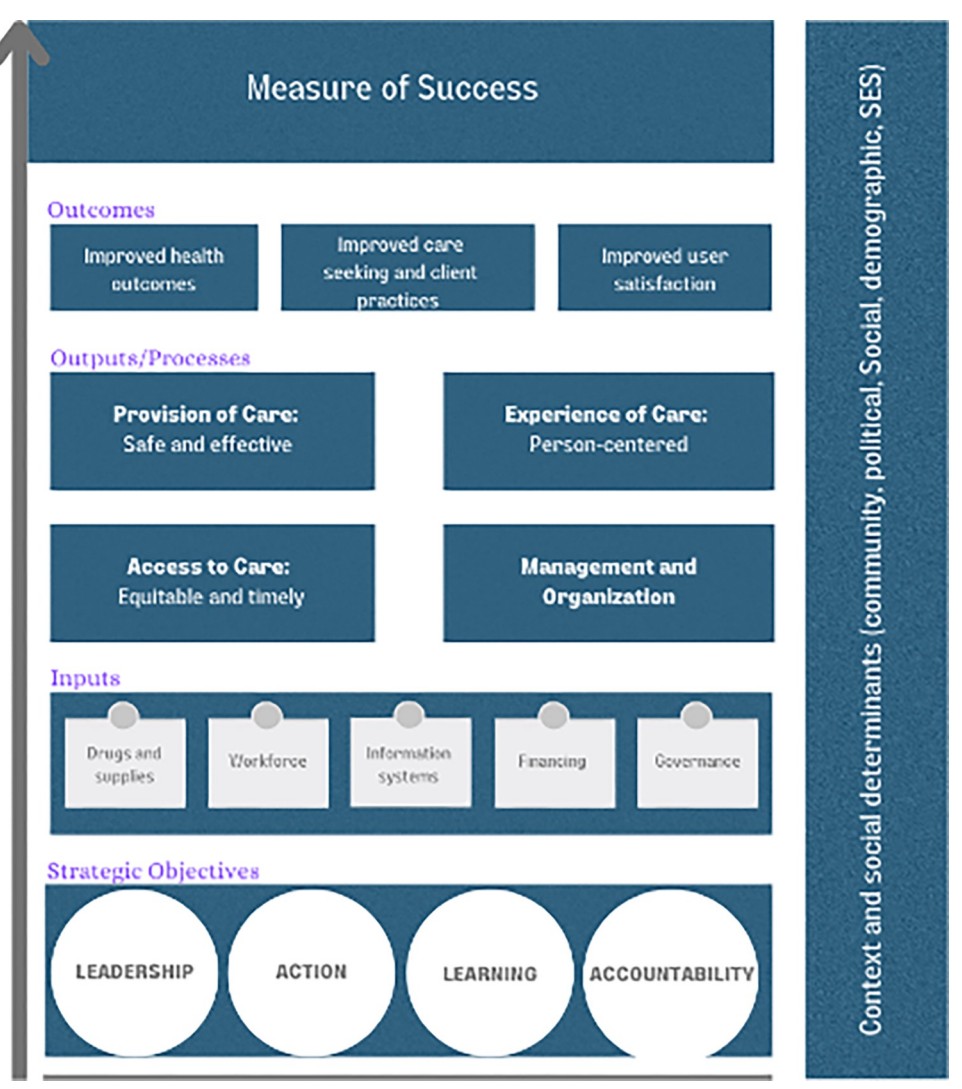

**Fig 1. Adaption of 'monitoring framework of quality equity dignity' from the WHO.**

framework is built on the achievements of other initiatives, such as EPMM, Every Newborn Action Plan (ENAP), and Global Strategy for Women's, Children's, and Adolescents' Health. It considers the needs and data uses at multiple levels–facility, district, national and global.

The framework followed the classic inputs, outputs/processes, outcomes framework, and it linked the strategic objectives (e.g., leadership, action, learning, and accountability) to the global goal of reducing maternal and newborn mortality. For inputs, it uses the WHO's health system's building blocks, including governance, health financing, information systems, workforce, and drugs and supplies. The framework focuses on health outcomes (e.g., maternal and child mortalities), care-seeking and client practice, and consumer satisfaction. In addition, it acknowledges that each country can customize the framework as needed for its unique context or priorities. However, it is recommended that each country should attempt to capture at least some indicators from the following four central elements shown in the framework: (a) management and organization; (b) access to care; (c) provision of care; and (d) experience of care.

The framework has eight domains of quality of care to measure processes. The health system provides structural inputs for quality improvement. Provision of care includes the use of evidence-based practices for routine and emergency care, information systems that store indicators for analysis and review, and systems for referral between different levels of care. Experience of care consists of effective communication with patients and their families about the care provided, their expectations and their rights; care with respect and preservation of dignity; and access to the social and emotional support of their choice. The cross-cutting areas of the framework include the availability of competent, motivated human resources and any necessary physical resources for health facilities. The eight components of the measurements and associated sub-components are shown in Table A in S3 Appendix.

Fig 2 shows an adaptation of the accountability/monitoring framework for the Countdown to 2030. The Countdown to 2030 is a global consortium aiming to strengthen the measurement and monitoring of RMNCH care and adolescent and nutrition care (RMNCAH+N). It also aims to enhance capacity for analysis and use of data at the country level. The Countdown's work has been concentrated on improving the measurement of coverage, quality of care and equity of RMNCAH+N. As its purpose is primarily to track the progress of relevant programs, quality of care is less of a focus. Besides the measurement of the general coverage

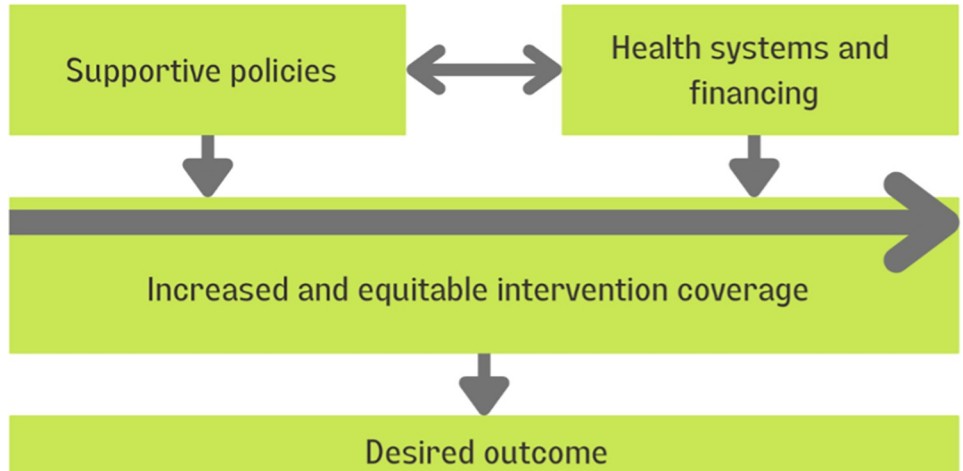

**Fig 2. Adaptation of 'Countdown to 2030 evaluation framework'.**

information of RMNCAH+N, the Countdown to 2030 also pays great attention to potential drivers to improve the coverage and quality of RMNCAH+N, which include indicators on supportive policies for health systems and financing. The indicators measuring supportive policies often concern the availability of RMNCAH+N policies and treatment protocols/guidelines while the indicators on health system and financing measure both financial and non-financial inputs to deliver RMNCAH+N services. In terms of specific RMNCH services, the Countdown to 2030 includes services spanning from the pre-pregnancy period to childhood.

Fig 3 shows the accountability/monitoring framework for the EWEC's Global Strategy. A commitment to the principles of survive, thrive and transform drives the Global Strategy, with the goal of ending preventable mortality and enabling women, children and adolescents to enjoy good health while contributing to transformative changes and sustainable development [15]. To ensure effective monitoring, the strategy aims to leverage existing SDG monitoring mechanisms to track the progress of country RMNCH programs towards their individual targets and the SDGs. Consistent with the three driving forces of the Global Strategy, there are 16 key monitoring indicators measuring survival outcomes (e.g., maternal and infant mortality rate), thriving status (e.g., assurance of health and wellbeing), and transformative environments (e.g., enabling environments such as health system and education) related to providing RMNCH services. With targeted measurements, the strategy was intended to strengthen countries' capacity to analyze, disseminate, and use collected data for country-led decision-making related to implementation and accountability. The strategy treated monitoring as a starting point of the accountability that encompasses monitoring, reviewing, and acting [15].

**Indicators analysis results.** The QED framework has 158 indicators (Table 1). Among them, there are quality improvement measures for health facilities, which are meant to support rapid improvements in quality of care led by facility quality improvement teams. There are also district/regional performance measures that are often used to hold health administrators accountable [13]. Table 1 shows the characteristics of quality indicators related to RMNCH services extracted from the QED framework. Given that this framework focuses on maternal and newborn health, there were no indicators on child and reproductive health services. Among all 158 indicators, input and output/process indicators account for many of the indicators. On the domain of indicators, we found that there was no efficiency indicator. The framework indicated that the equity domain crosses the spectrum of indicators, and thus no specific equity indicator was expected. Most indicators are effectiveness indicators or safety indicators, which account for 37.3% and 24.7% of the total number of indicators, respectively. There are 24 indicators which are difficult to categorize. The detailed indicators and their categorizations could be found in Table B in S3 Appendix.

There is a total of 60 non-duplicated RMNCH indicators from the Countdown to 2030, Global Strategy, and EWEC monitoring frameworks (Table 2). Of them, 13 indicators focus on general quality (mostly on the inputs for delivering RMNCH services, such as human resources), 4 on reproductive health, 19 on maternal health, 4 on newborn health, and 20 on child health.

The output/process indicators represent the largest share of quality indicators. There are about the same number of input indicators as outcome indicators. As to the domain of indicators, the effectiveness indicators were most prevalent, but there are a limited number of indicators on safety, timeliness, and patients' responsiveness. There was no indicator measuring the efficiency of care. Across different health services, the distribution of domains of indicators was similar. The detailed indicators and their categorizations could be found in Table C in S3 Appendix.

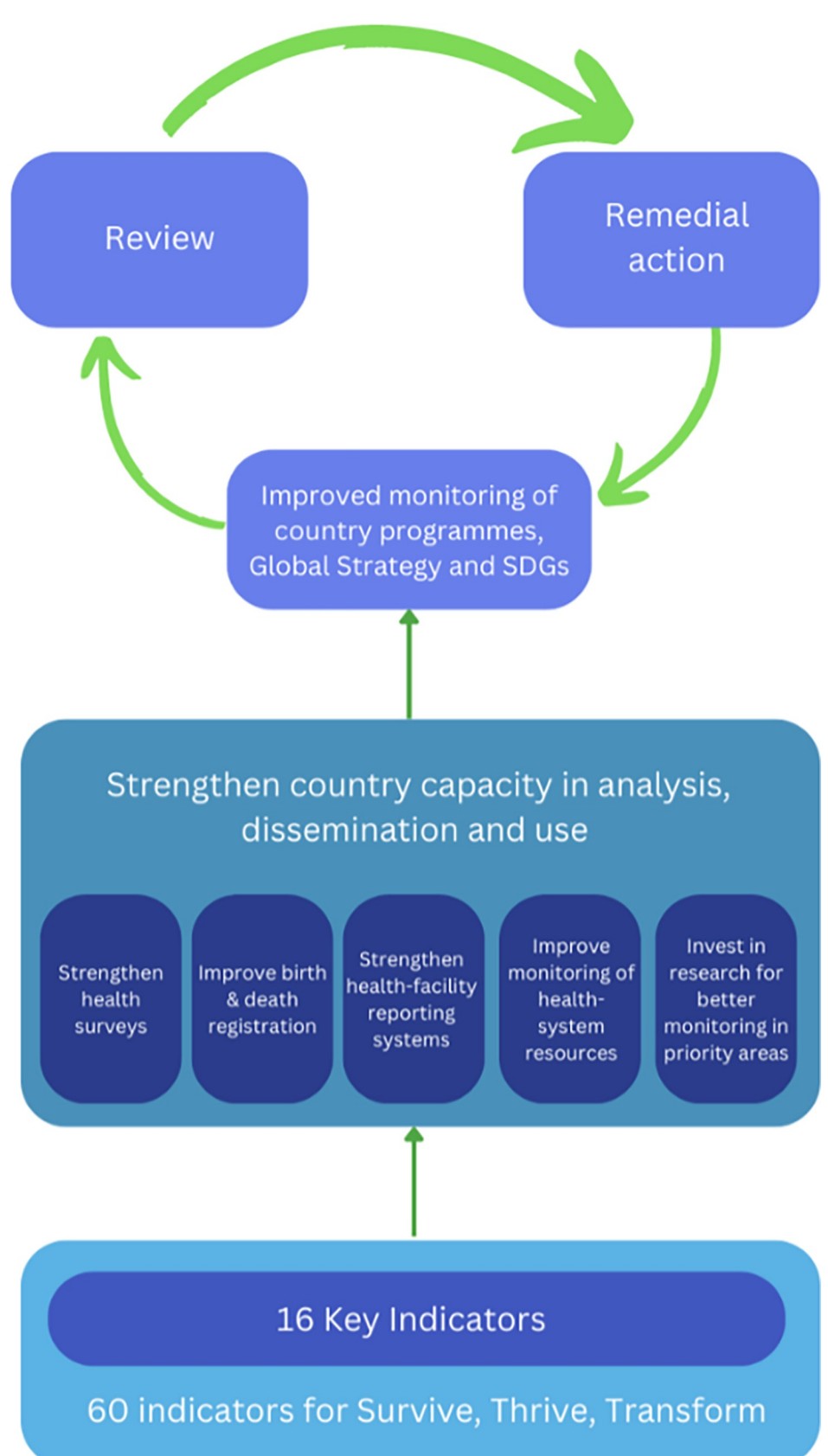

**Fig 3. Adaptation of 'framework used by the Global Strategy'.**

**Table 1. Characteristics of quality indicators in QED framework by type of services.**

| Type of quality indicators | General | Maternal | Newborn | Total |
|---|---|---|---|---|
| Input | 38(62.3%) | 13(18.6%) | 7(25.9%) | 58(36.7%) |
| Output/process | 21(34.4%) | 36(51.4%) | 12(44.4%) | 69(43.7%) |
| Outcome | 2(3.3%) | 21(30.0%) | 8(29.6%) | 31(19.6%) |
| Domain of indicators | | | | |
| Efficiency | (0.0%) | (0.0%) | (0.0%) | (0.0%) |
| Effectiveness | 10(16.4%) | 33(47.1%) | 16(59.3%) | 59(37.3%) |
| Safety | 11(18.0%) | 21(30.0%) | 7(25.9%) | 39(24.7%) |
| Timeliness | 6(9.8%) | 6(8.6%) | 3(11.1%) | 15(9.5%) |
| Patient-centered | 12(19.7%) | 9(12.9%) | 0(0.0%) | 21(13.3%) |
| uncategorized | 22(36.1%) | 1(1.4%) | 1(3.7%) | 24(15.2%) |
| Monitoring level | | | | |
| Health facility | 15(24.6%) | 57(81.4%) | 20(74.1%) | 92(58.2%) |
| District/regional | 46(75.4%) | 13(18.6%) | 7(25.9%) | 66(41.8%) |
| Data collection | | | | |
| RIS | 5(8.2%) | 34(48.6%) | 16(59.3%) | 55(34.8%) |
| client interview | 2(3.3%) | 12(17.1%) | 1(3.7%) | 15(9.5%) |
| observation | 23(37.7%) | 7(10.0%) | 2(7.4%) | 32(20.3%) |
| others | 9(14.8%) | 0(0.0%) | 0(0.0%) | 9(5.7%) |
| mixed | 22(36.1%) | 17(24.3%) | 8(29.6%) | 47(29.8%) |

Note: RIS denotes routine information system

## Discussion

### Statement of principal findings

Developing quality indicators and monitoring them are important elements of accountability, and health systems blocks are critical inputs for QoC. Thus, it is important to measure quality through both condition-specific indicators and general health system indicators. However, given the different purposes of global monitoring initiatives, the indicators used by the global initiatives vary substantially. There is a lack of indicators measuring QoC of reproductive health. In terms of quality domains, the timeliness and efficiency of RMNCH services are often neglected.

**Table 2. Characteristics of quality indicators in the Countdown to 2030, Global Strategy, and Every Women Every Child monitoring framework.**

| Characteristics | General | Reproductive | Maternal | Newborn | Child | Total |
|---|---|---|---|---|---|---|
| Type of quality indicators | | | | | | |
| Input | 11 (84.6%) | 1 (25.0%) | 1 (5.3%) | 0 (0.0%) | 0 (0.0%) | 13 (21.7%) |
| Output/process | 2 (15.4%) | 2 (50.0%) | 11 (57.9%) | 2 (50.0%) | 16 (80.0%) | 33 (55.0%) |
| Outcome | 0 (0.0%) | 1 (25.0%) | 7 (36.8%) | 2 (50.0%) | 4 (20.0%) | 14 (23.3%) |
| Domain of indicators | | | | | | |
| Efficiency | 0 (0.0%) | 0 (0.0%) | 0 (0.0%) | 0 (0.0%) | 0 (0.0%) | 0 (0.0%) |
| Effectiveness | 2 (15.4%) | 2 (50.0%) | 16 (84.2%) | 4 (100.0%) | 16 (80.0%) | 40 (66.7%) |
| Safety | 1 (7.7%) | 0 (0.0%) | 1 (5.3%) | 0 (0.0%) | 1 (5.0%) | 3 (5.0%) |
| Timeliness | 0 (0.0%) | 0 (0.0%) | 1 (5.3%) | 0 (0.0%) | 2 (10.0%) | 3 (5.0%) |
| Patient-centered | 0 (0.0%) | 1 (25.0%) | 0 (0.0%) | 0 (0.0%) | 0 (0.0%) | 1 (1.7%) |
| uncategorized | 10 (76.9%) | 1 (25.0%) | 1 (5.3%) | 0 (0.0%) | 1 (5.0%) | 13 (21.7%) |
| Total | 13 (100.0%) | 4 (100.0%) | 19 (100.0%) | 4 (100.0%) | 20 (100.0%) | 60 (100%) |

## Strengths and limitations

A few limitations of this study should be acknowledged. Firstly, we only reviewed accountability/monitoring frameworks from a few global monitoring initiatives and did not conduct a systematic review of frameworks within countries. Few countries share their accountability/monitoring frameworks at the national level with detailed QoC indicators. Secondly, the review focused on measurements and monitoring, and less on remedial action to correct quality concerns, although the latter is an important element of accountability. Thirdly, the analysis of the indicators from the global monitoring initiatives is limited by the purpose of each of these initiatives; consequently, monitoring indicators at the facility level did not fall within the scope of most of these initiatives. Despite these limitations, the review provides useful information for developing monitoring and accountability indicators.

## Interpretation within the context of the wider literature

One of the common themes emerging from the study is that monitoring RMNCH indicators is an important component of accountability. The accountability strategies to improve QoC vary, and countries may take different strategies to improve accountability for QoC. However, each of the approaches relies on developing clearly defined and measurable QoC indicators to demonstrate that health services are provided with assured quality, which makes monitoring a key component of accountability. In fact, accountability goes beyond developing and monitoring QoC indicators and includes reporting provider quality and incentivizing providers [20]. The Global Strategy uses three components of accountability: monitoring, reviewing, and acting.

We found that the accountability/monitoring frameworks do not provide detailed information on how indicators are matched to different quality assurance or improvement mechanisms. There are many mechanisms to increase accountability, including training and coaching, auditing, accreditation, public reporting, and continuous quality improvement (CQI). Different mechanisms require indicators with different features. For example, accreditation interventions require that a set of indicators is reviewed by an independent agency at the facility level in a short time. It generally requires that the indicators be comprehensive, cover a wide range of services, and can be quickly assembled using routine information systems. In contrast, CQI is often targeted toward specific areas for improvement with a narrower scope of indicators. To allow for the smooth operation of the accountability/monitoring framework, it would be useful to provide guidelines to develop different sets of indicators for different QI mechanisms.

Additionally, we found that the existing indicators are less focused on timeliness. Timeliness is an important element of quality of care. This is particularly important in LMICs where the delay of treatment or seeking care is common. Even in the United States, it was estimated that 3 out of 10 Americans delayed care, primarily due to financial reasons [21]. The economic cost of delaying and forgoing needed treatment is high, amounting to $39.5 billion in the United States alone in 2013; it constitutes the second largest amount of loss in the six opportunity areas to save resources [22]. Among the frameworks that we reviewed, only the QED framework contains indicators on timeliness with a focus on referrals. Given the importance of timeliness in preventing avoidable loss of life, more effort should be given to it to avoid potential economic loss and save lives.

## Implications for policy, practice, and research

All three frameworks take a health system approach to addressing QoC concerns. The WHO's building block framework is embedded in all monitoring frameworks, although the roles of

health system structure within the framework vary. In the QED framework, the health system structure serves as an input to improve QoC. The Countdown to 2030 and Global Strategy frameworks regard health system components, such as policy and financing, as environmental indicators. Therefore, health providers must be held accountable for non-service specific procedures and outcomes in addition to those that are service-specific. Non-service specific indicators may include the existence of relevant policies and regulations and availability of financial resources. This is consistent with the requirement that the accountability mechanisms for QoC of RMNCH should hold providers not only clinically accountable but also operationally accountable. To ensure the relevance of these indicators to QoC, conducting a baseline QoC assessment is often suggested to understand where QoC concerns lie in the country of interest.

The frameworks we reviewed seem more focused on public-sector health facilities. There is little guidance on reporting from the private sector, and on how to hold the private sector more accountable, particularly for health facilities where health information systems may not be available. Countries should implement national mandatory reporting indicators and incorporate key quality indicators in their national health information system. Special attention and technical support, if needed, should be given to health facilities with poor information systems and to the private sector for data reporting and reviewing. This will ensure data reporting is congruent with national and international standards.

## Conclusion

Global monitoring initiatives provide valuable frameworks for countries to understand which key indicators need to be tracked to achieve global objectives and to develop the foundation for their own accountability/monitoring systems. However, there are gaps in quality indicator design and use. Countries need to build on what the global initiatives have achieved to systematically examine their quality concerns, develop a more tailored and effective accountability/monitoring framework, and improve population health.

## Supporting information

**S1 Appendix. Global monitoring initiatives included in the review.**
(DOCX)

**S2 Appendix. Definition of the six dimensions of quality of care.**
(DOCX)

**S3 Appendix. Quality measures in global monitoring initiative frameworks.**
(DOCX)

## Author Contributions

**Conceptualization:** Eva Jarawan, Wu Zeng.

**Formal analysis:** Eva Jarawan, Mara Boiangiu, Wu Zeng.

**Supervision:** Wu Zeng.

**Writing – original draft:** Wu Zeng.

**Writing – review & editing:** Eva Jarawan, Mara Boiangiu, Wu Zeng.

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
