## [Decision Letter · Decision Letter 0]

7 Aug 2023

PGPH-D-23-00527

Strengthening provider accountability: A scoping review of accountability/monitoring frameworks for quality of RMNCH care

Dear Dr. Zeng,

Thank you for submitting your manuscript to PLOS Global Public Health. After careful consideration, we feel that it has merit but does not fully meet PLOS Global Public Health’s publication criteria as it currently stands. Therefore, we invite you to submit a revised version of the manuscript that addresses the points raised during the review process.

Two reviewers have provided constructive feedback below.

Please ensure that your decision is justified on PLOS Global Public Health’s publication criteria and not, for example, on novelty or perceived impact.

We look forward to receiving your revised manuscript.

Kind regards,

Hannah Tappis, DrPH, MPH

Academic Editor

Journal Requirements:

Additional Editor Comments (if provided):

Reviewers' comments:

Reviewer's Responses to Questions

**Comments to the Author**

1. Does this manuscript meet PLOS Global Public Health’s publication criteria? Is the manuscript technically sound, and do the data support the conclusions? The manuscript must describe methodologically and ethically rigorous research with conclusions that are appropriately drawn based on the data presented.

Reviewer #1: Yes

Reviewer #2: Partly

2. Has the statistical analysis been performed appropriately and rigorously?

Reviewer #1: Yes

Reviewer #2: N/A

3. Have the authors made all data underlying the findings in their manuscript fully available (please refer to the Data Availability Statement at the start of the manuscript PDF file)?

Reviewer #1: Yes

Reviewer #2: Yes

4. Is the manuscript presented in an intelligible fashion and written in standard English?

Reviewer #1: Yes

Reviewer #2: Yes

5. Review Comments to the Author

Reviewer #1: The authors did a good work on reviewing the existing literature on monitoring frameworks for RMNCH. Methods for identifying and assessing the frameworks is clear and seems fit for purpose. The identified frameworks are well summarised and assessed clearly. Results are clear and easy to read. Conclusions are according to findings. If a country is looking into developing a monitoring framework for RMNCH services this paper does a good job in summarising and describing what is available.

The paper also includes some case studies that are mentiones in the introduction and methods, but not presented in the results nor in the discussion. The annex with the case studies is quite bland and does not add value to the rest of the article. Authors should seriously consider removing this part and probably use it in anothe publication.

Reviewer #2: This paper has promise, but needs to be strengthened in a few places.

The authors argue that Quality of Care is a ‘method’--- more needs to be added to this assertion and the evidence base that support QoC as an approach.

The authors make an important link between QofC and accountability but the reviewer suggests using references related to accountability in the health space as opposed to what they have there. Suggest adding more language about why accountability is an important component that is underpinning QofC. There are some important pieces out there on this that should be referenced.

Methods related comments:

Authors should define global monitoring initiatives and clarify selection criteria. There needs more explanation on why/ how those specific ones were chosen.

Suggest that authors explain/justify why scoping approach was used versus systematic review, and add in the definition/criteria for using scoping review methodology.

In search approach: Use of the word quick (quick review, quick search) does not provide enough information on exclusion/inclusion criteria.

6. PLOS authors have the option to publish the peer review history of their article (what does this mean?). If published, this will include your full peer review and any attached files.

**Do you want your identity to be public for this peer review?** For information about this choice, including consent withdrawal, please see our Privacy Policy.

Reviewer #1: No

Reviewer #2: No

---

## [Decision Letter · Decision Letter 1]

19 Oct 2023

Strengthening provider accountability: A scoping review of accountability/monitoring frameworks for quality of RMNCH care

PGPH-D-23-00527R1

Dear Dr. Zeng,

We are pleased to inform you that your manuscript 'Strengthening provider accountability: A scoping review of accountability/monitoring frameworks for quality of RMNCH care' has been provisionally accepted for publication in PLOS Global Public Health.

Best regards,

Hannah Tappis, DrPH, MPH

Academic Editor

Reviewer Comments (if any, and for reference):

Reviewer's Responses to Questions

**Comments to the Author**

1. If the authors have adequately addressed your comments raised in a previous round of review and you feel that this manuscript is now acceptable for publication, you may indicate that here to bypass the “Comments to the Author” section, enter your conflict of interest statement in the “Confidential to Editor” section, and submit your "Accept" recommendation.

Reviewer #2: All comments have been addressed

2. Does this manuscript meet PLOS Global Public Health’s publication criteria? Is the manuscript technically sound, and do the data support the conclusions? The manuscript must describe methodologically and ethically rigorous research with conclusions that are appropriately drawn based on the data presented.

Reviewer #2: Yes

3. Has the statistical analysis been performed appropriately and rigorously?

Reviewer #2: N/A

4. Have the authors made all data underlying the findings in their manuscript fully available (please refer to the Data Availability Statement at the start of the manuscript PDF file)?

Reviewer #2: Yes

5. Is the manuscript presented in an intelligible fashion and written in standard English?

Reviewer #2: Yes

6. Review Comments to the Author

Reviewer #2: This looks good and ready for publication. I assume that a copy edit will take place? there are a few places where where the word didn't is used instead of did not. I would also suggest removing or clarifying the use of the following words:

significant (page 7)

slightly and pure (page 8)

on page 8 you mention patient centered and person centered-- would suggest choosing one and using it.

on page 12-- more than 158? or are there 158 indicators?

Thank you!

7. PLOS authors have the option to publish the peer review history of their article (what does this mean?). If published, this will include your full peer review and any attached files.

**Do you want your identity to be public for this peer review?** For information about this choice, including consent withdrawal, please see our Privacy Policy.

Reviewer #2: No
